# Characterization of PcSTT3B as a Key Oligosaccharyltransferase Subunit Involved in N-glycosylation and Its Role in Development and Pathogenicity of *Phytophthora capsici*

**DOI:** 10.3390/ijms24087500

**Published:** 2023-04-19

**Authors:** Tongshan Cui, Quanhe Ma, Fan Zhang, Shanshan Chen, Can Zhang, Jianjun Hao, Xili Liu

**Affiliations:** 1Department of Plant Pathology, College of Plant Protection, China Agricultural University, Beijing 100193, China; cuitongshan0619@163.com (T.C.); maqh191@163.com (Q.M.); fan_zhang@cau.edu.cn (F.Z.); 18800172207@163.com (S.C.); zhangcanaimama@163.com (C.Z.); 2School of Food and Agriculture, University of Maine, Orono, ME 04469, USA; jianjun.hao1@maine.edu; 3State Key Laboratory of Crop Stress Biology for Arid Areas, College of Plant Protection, Northwest A&F University, Yangling 712100, China

**Keywords:** N-glycosylation, oligosaccharyltransferase, PcSTT3B, virulence, *Phytophthora capsici*

## Abstract

Asparagine (Asn, N)-linked glycosylation is a conserved process and an essential post-translational modification that occurs on the NXT/S motif of the nascent polypeptides in endoplasmic reticulum (ER). The mechanism of N-glycosylation and biological functions of key catalytic enzymes involved in this process are rarely documented for oomycetes. In this study, an N-glycosylation inhibitor tunicamycin (TM) hampered the mycelial growth, sporangial release, and zoospore production of *Phytophthora capsici*, indicating that N-glycosylation was crucial for oomycete growth development. Among the key catalytic enzymes involved in N-glycosylation, the *PcSTT3B* gene was characterized by its functions in *P. capsici*. As a core subunit of the oligosaccharyltransferase (OST) complex, the staurosporine and temperature sensive 3B (STT3B) subunit were critical for the catalytic activity of OST. The *PcSTT3B* gene has catalytic activity and is highly conservative in *P. capsici*. By using a CRISPR/Cas9-mediated gene replacement system to delete the *PcSTT3B* gene, the transformants impaired mycelial growth, sporangial release, zoospore production, and virulence. The *PcSTT3B*-deleted transformants were more sensitive to an ER stress inducer TM and display low glycoprotein content in the mycelia, suggesting that PcSTT3B was associated with ER stress responses and N-glycosylation. Therefore, PcSTT3B was involved in the development, pathogenicity, and N-glycosylation of *P. capsici*.

## 1. Introduction

The oomycetes are a class of eukaryotic microorganisms that have similar life cycles and growth habits to the filamentous fungi. Oomycotes contains many pathogens of both plants and animals [1,2], including the infamous *Phytophthora infestans* that was responsible for the Irish potato famine of the nineteenth century and which remains a severe threat to potato production to this day [3]. In most instances, the oomycetes differ from fungi in many ways, including their genome size, ploidy of vegetative hyphae, cell wall composition (cellulose instead of chitin), and the type of mating hormones produced [4]. *Phytophthora capsici* is among the top five most important plant pathogens, with a wide host range, including species from *Solanaceae*, *Leguminosae*, and *Cucurbitaceae*. It is estimated that the worldwide vegetable production, valued at over one billion dollars, is threatened by *P. capsici* alone each year [5,6]. Its life cycle is divided into sexual and asexual reproductions. In the sexual life cycle, *P. capsici* can produce long-lived dormant oospores, which can be used as the main source of preliminary infection in soil. *P. capsici* is heterothallic and thus needs two strains with different mating types for sexual reproduction. At the asexual stage of the life cycle, *P. capsici* produces sporangia. Mature sporangia release zoospores under low temperatures and free water conditions. Released zoospores form cysts after reaching the host surface. The cyst quickly produces a germ tube and forms hyphae to invade the epidermis of the host plant or colonize its tissue [7,8,9]. Thus, the zoospore is a key stage for the infection of *P. capsici* [5]. Since *P. capsici* infects a wide variety of crops and is difficult to be controlled, developing a promising strategy for the development of fungicides with new modes of action is critical.

N-glycosylation is one of the post-translational modifications in eukaryotes. N-glycosylation regulates many important biological processes, including cell differentiation, growth and development, and intercellular communication [10]. It affects protein folding, stability, and localization [11]. N-glycosylation occurs in the endoplasmic reticulum (ER), in the Golgi apparatus (GA), and in the secretory system. Initially, a lipid-linked oligosaccharide (LLO, Glc_3_Man_9_GlcNac_2_) is biosynthesized via adding two N-acetylglucosamines, nine mannoses, and three glucoses to phosphorylated dolichol [10]. Tunicamycin (TM) blocks the function of dolichyl-phosphate N-acetylglucosamine phosphotransferase 1 (DPAGT1), thereby preventing the synthesis of all glycan precursors and N-linked glycosylation [12,13]. Subsequently, the oligosaccharyltransferase (OST) transfers the LLO to Asn residues in the NXS/T (Asn-X-Ser/Thr, where X any amino acid except Pro) motifs within nascent polypeptides [14,15]. Finally, N-glycan trimming removes terminal glucose and mannose residues via the glucosidases I (GCSI), glucosidase II (GCSII) and mannosidases I (MAN I) [16,17]. Proteins are added to complex N-glycans in the GA [11]. Because of the important catalytic role of OST in N-glycosylation, its catalytic mechanism has been widely concerned and studied in various species.

In eukaryotes, OST composes a catalytic subunit, staurosporine and temperature sensive 3 (STT3), and other auxiliary subunits [18,19]. In mammals, the STT3A isoform interacts with the protein translocation channel (Sec61 complex) and is involved in cotranslational glycosylation by an N-terminal to C-terminal scanning mechanism. The STT3B isoform participates in posttranslational glycosylation, which has been skipped by STT3A, including extreme C-terminal sites [20,21]. In *Arabidopsis thaliana*, STT3A and STT3B are homologous to the STT3 of yeasts and mammals. In the Koiwa et al. (2003) study, the *AtSTT3A* and *AtSTT3B* double knockout results in gametophytic lethality [22]. *STT3A*-deficiented plant is viable but have altered the biogenesis of heavily glycosylated proteins. In *Saccharomyces cerevisiae*, the OST complex comprises a single conserved STT3 protein and seven additional non-catalytic subunits that contribute to the N-glycosylation by regulating substrate specificity, complex stability, and complex assembly. STT3, OST1, OST2, WBP1, and SWP1 are essential for cell viability [14,15]. In the human pathogen *Aspergillus fumigatus*, researchers failed to delete the *AfSTT3* gene and repressed it, leading to severe retardation of growth by replacing its endogenous promoter [23]. In the plant pathogen *Verticillium dahliae*, STT3 is involved in fungal growth, glycoprotein secretion, and virulence [24]. However, the roles of the OST subunit STT3 in plant pathogenic oomycetes are still unclear, especially in growth development and oomycete-host plant interactions.

The objectives of this study were to (1) examine the impact of TM and Dithiothreitol (DTT) on mycelial growth, sporangial release, and zoospore production in *P. capsici*, (2) determine the functions of PcSTT3B in N-glycosylation, and (3) characterize the mannosylated glycoproteins in responses to the deletion of the *PcSTT3B* gene.

## 2. Results

### 2.1. The Inhibition of TM and DTT in P. capsici

To determine the functions of N-glycosylation in *P. capsici*, the effects of two well-known ER stress inducers (TM and DTT) were initially investigated on the growth and development of *P. capsici*. TM is an N-glycosylation inhibitor, which blocks LLO formation [13,25]. DTT is a reducing agent, which prevents the formation of disulfide bonds and forms misfolded proteins in the ER [25,26]. The mycelial growth, sporangial release and zoospore production were significantly decreased when treated with TM (0.4 µg/mL or 0.7 µg/mL) or DTT (6 mM or 8 mM), compared to the untreated control (CK, without ER stress inducers) (Figure 1A,B,D,E). TM and DTT had no significant difference on the sporangium production (Figure 1C). Adding TM or DTT significantly increased cyst germination rates compared to the CK (Figure 1A,F). These results demonstrated that N-glycosylation likely played an essential role in the growth and development of *P. capsici*.

### 2.2. Sequence and Phylogenetic Analysis of PcSTT3B

As the key catalytic subunit of OST, the sequence of the core subunit STT3s of *P. capsici* was characterized. The genes *PcSTT3A* (JGI accession: Pc118787) and *PcSTT3B* (JGI accession: Pc541727) were identified in the JGI database [27] in the *P. capsici* genome. They have a high similarity to the STT3s in other *Phytophthora* species. *PcSTT3A* and *PcSTT3B* encode the STT3/PMT2 superfamily domain [28] (Figure 2A). Based on Sanger sequencing with cDNA and DNA of the wild-type (WT), the *PcSTT3B* gene contained three introns (59, 60, and 62 bp) and a 2208 bp open reading frame (ORF) encoding a 735-amino-acid protein to correct the *PcSTT3B* gene model (Appendix A).

Phylogenetic analysis showed that most oomycetes contained two distinct STT3 isoforms, STT3A and STT3B, which are grouped into one branch. However, fungi included only one homologous STT3 protein. The PcSTT3B was highly conserved within oomycetes (Figure 2A). Sequence alignment (Figure 2A) suggested that PcSTT3B comprised a highly canonical WWDYG motif, which acted as the enzyme activity pocket and provided a hydrogen bond to interact with the Thr/Ser of the NXT/S motif [14,29]. The PcSTT3B protein carried the DXXK motif that contributed additional contacts to the Thr/Ser of the acceptor peptide [14]. Furthermore, we identified 11 transmembrane domains in PcSTT3B (Figure 2B), implying it was probably a transmembrane protein.

### 2.3. Transcription Profile of the PcSTT3B Gene

To investigate the potential roles of the *PcSTT3B* gene, we analyzed expression levels of the *PcSTT3B* gene in the development and infection stages, including mycelia (MY), sporangia (SP), zoospores (ZO), germinated cysts (CY), and infection stages (0, 1.5, 3, 6, 12, 24, and 48 h after infection in the susceptible pepper leaves). The reverse transcription quantitative real-time polymerase chain reaction (RT-qPCR) showed that the *PcSTT3B* gene was expressed in all stages as described above. In the growth and development stages, the expression levels of the *PcSTT3B* gene display no significant differences in the MY, SP, ZO, or CY stages (Figure 3A). The expression levels of the *PcSTT3B* gene were significantly upregulated after infection for 3, 6 and 24 h, compared to the stage of 0 h (Figure 3B). These results indicated that the *PcSTT3B* gene might function prominently in the pathogenic stage of *P. capsici*.

### 2.4. PcSTT3B Gene Disruption and Complementation

To investigate the biological functions of *PcSTT3B* in *P. capsici*, homology-directed repair (HDR)-mediated replacement of the *PcSTT3B* gene with an *NPTII* gene was conducted in the *P. capsici* genome using CRISPR/Cas9 [30,31]. After G418 resistance screening, *PsSTT3B*-deleted transformants (B8, B146) were examined by PCR and RT-qPCR analysis. The transformant, which was transformed but failed to delete the *PcSTT3B* gene, was regarded as a control strain (B-CK). The transformant CB151 was complemented by reintroducing *PcSTT3B* through the CRISPR/Cas9 mediated in situ complementation method [32] (Figure 4A). PCR analysis indicated the presence of the exogenous *NPTII* gene and the absence of the *PcSTT3B* gene in the *PsSTT3B*-deleted transformants (Figure 4B). Additionally, RT-qPCR analysis showed that the expression of the *PcSTT3B* was not detected in the RT-qPCR assay, whereas the expression of the *PcSTT3A* gene was significantly up-regulated in the mycelia stage of the *PcSTT3B*-deleted transformants compared to the WT strain BYA5 (Figure 4C). These results indicated that the expression patterns of the *PcSTT3B* and *PcSTT3A* genes might be complementary.

### 2.5. Biological Characteristics of the PcSTT3B-Deleted Transformants

To further investigate the roles of PcSTT3B in the growth and development of *P. capsici*, biological characteristics of the BYA5, B-CK, CB151, and the three *PcSTT3B*-deleted transformants were examined. Mycelium growth, sporangial release rate and zoospore production of the *PcSTT3B*-deleted transformants were significantly lower compared to the BYA5, B-CK, and CB151 strains (Figure 5A,C,D). The BYA5, B-CK, CB151, and the *PcSTT3B*-deleted transformants displayed no significant differences in the sporangium production and cyst germination rates (Figure 5A,B,E). These results indicated that *PcSTT3B* was involved in the mycelial growth, sporangial release rate, and zoospore production.

When *P. capsici* colonized pepper, the expression levels of the *PcSTT3B* gene were highly upregulated to varying degrees, implying that *PcSTT3B* may play an important role in the infection stage. To explore whether *PcSTT3B* was involved in pathogenicity, lesion diameters, caused by the *PcSTT3B*-deleted transformants, were significantly lower than that of the BYA5, B-CK, and CB151 strains (Figure 6A,B). The result demonstrated that *PcSTT3B* was involved in the zoospores-infection of *P. capsici*.

### 2.6. Responses of PcSTT3B to ER Stress in P. capsici

Since the absence of N-glycosylation may disrupt ER homeostasis, two well-characterized ER stress inducers, TM and DTT, were used to test the effect of PcSTT3B on the BYA5, B-CK, CB151, and *PcSTT3B*-deleted transformants. BYA5, B-CK, and *PcSTT3B*-deleted transformants were grown on V8 agar plates supplemented with 0.5 µg/mL TM or 6 mM DTT. All strains were inhibited by TM and DTT. The *PcSTT3B*-deleted transformants were more sensitive to TM compared to the WT, B-CK, or CB151 strains. However, there were no significant changes in the inhibition rates of all strains by adding DTT (Figure 7A,B). The *PcSTT3B* transcript levels increased significantly after TM treatment, instead of DTT (Figure 7D). Therefore, these results indicated that *PcSTT3B* might be responsible for the ER stress tolerance caused by TM in *P. capsici*.

### 2.7. Disruption of PcSTT3B Abolished the Attachment of Glycoproteins in Mycelium

Because PsSTT3B is a highly conserved catalytic subunit, we were interested in investigating whether the deletion of the *PcSTT3B* gene affected the glycoprotein content. The lectin ConA recognizes and binds to oligomannosidic glycans and had a high affinity to both terminal α-D-mannose and α-D-glucose residues of glycoproteins [33]. Certain mannosylated protein bands (red arrow) were slightly reduced in the *PcSTT3B*-deleted transformants compared to that of the BYA5 strain (Figure 7C). The results demonstrated that deletion of the *PsSTT3B* gene decreased the high mannose-type sugar chains in some proteins in *P. capsici*.

## 3. Discussion

N-glycosylation within the N-X-Ser/Thr (X ≠ Pro) motif is a post-translational modification that is ubiquitous and participates in various cellular processes. In *P. sojae*, the N-glycosylation inhibitor at 2.5 μg/mL of TM prevents the growth of *P. sojae* and cyst germination [34]. In *Magnaporthe oryzae*, 5 μg/mL of TM significantly inhibited mycelial growth, mycelial cell length, conidiophore formation, and invasive hyphal growth in host cells [26]. Similarly, TM and DTT prevented mycelial growth, sporangial release and zoospore production of *P. capsici*, suggesting that N-glycosylation is necessary for oomycete development. However, TM and DTT significantly increased the cyst germination rates, which might need to be maintained within a range that was consistent with the rates before the addition of TM and DTT to result in strong pathogenicity. Therefore, high germination rates, after the addition of TM or DTT, might not necessarily indicate strong pathogenicity.

The key catalytic enzyme OST is important for N-glycosylation. The eukaryotic OST complex is highly conserved, including the active center STT3 subunit and several non-catalysis subunits [14,15,18]. However, there are differences in the STT3 proteins among species. The prokaryotic OST enzyme contains a membrane-embedded single subunit: archaeal glycosylation B (AglB) for archaea, and protein glycosylation B (PglB) for bacteria, which are homologs of STT3 [35]. *Leishmania major* or *Trypanosoma brucei* have four or three STT3 copies, LmSTT3A-D or TbSTT3A-C. In other words, these unicellular parasites lack additional, non-catalytic subunit homologs of the yeast or mammal OST complex [36,37]. These single subunit OST (ssOST) enzymes display distinct protein acceptor and LLO donor specificity [38,39]. Most oomycetes utilize two distinct STT3 isoforms, similar to plants and mammals [18,40,41]. STT3A and STT3B of most oomycetes are grouped independently into one branch on the phylogenetic tree. Fungi have only an STT3 homologous protein, as do bacteria. OST3/6 interfaces with the Sec61 or Ssh1 translocon, which exists in two yeast OST complexes, to involve in cotranslational or posttranslational glycosylation in *S. cerevisiae* [14,15,42]. However, the DC2/KPC2 (Homologous with OST6)-containing STT3A-OST functions cotranslationally, and the MagT1/TUSC3 (Homologous with OST3)-containing STT3B-OST functions posttranslationally in the mammal [20,43]. Overall, the contributions of PcSTT3A or PcSTT3B to cotranslational or posttranslational glycosylation remain unclear and need further exploration.

The OST complex transfers the LLO from the lipid carrier phosphorylated dolichol to the NXS/T motif of a nascent polypeptide in higher eukaryotes [44,45]. The deletions of the five subunits (STT3, SWP1, WBP1, OST1, and OST2) are fatal, whereas deletions of other subunits (OST3, OST4, and OST5) reduce the complex stability and glycosylation activity in *S. cerevisiae* [46,47]. In *A. thaliana*, the *AtSTT3A* and *AtSTT3B* double knockout results in gametophytic lethality [22]. The situation is different among pathogenic fungi. *STT3* can be deleted in the genome of *V. dahliae*, whereas it cannot be deleted in *A. fumigatus* [23,24]. We generated the *PcSTT3B*-deleted transformants in *P. capsici* using the CRISPR/Cas9 system. The expression level of the *PcSTT3A* gene significantly increased in the *PcSTT3B*-deleted transformants. We speculate that *PcSTT3A* and *PcSTT3B* are redundant and collaborative. In *P. capsici*, *PcSTT3A* may partially supplement the function of *PcSTT3B*, so as to avoid cell death.

The STT3 has a catalytic active site and is conserved in the OST complex. The *AtSTT3A*-deleted mutant is more sensitive to salt, while the *AtSTT3B*-deleted mutant has no obvious phenotype in *A. thaliana* [22]. The *AtSTT3A* is involved in stomatal development by modulating the release of active abscisic acid and auxin [48]. In *A. fumigatus*, repression of the *AfSTT3* gene leads to severe retardation of growth and a slight defect in cell wall integrity by replacing the endogenous promoter of *AfSTT3* [23]. In *V. dahliae*, deletion of the *VdSTT3* gene results in defective growth, glycoprotein secretion, and virulence [24]. In this study, *PcSTT3B* was involved in mycelial growth, sporangial release, zoospore production, and virulence of *P. capsici*. The greater number of unreleased sporangia in the *PcSTT3B*-deleted transformants might have resulted in defecting zoospore production. The pathogenicity in the oomycete-plant interaction is affected by many aspects, such as mycelial growth rate, cyst germination, and formation of the infection structure [49,50,51]. Interestingly, the cyst germination rate and morphology were not affected, but the zoospore-inoculated pathogenicity was significantly reduced in the *PcSTT3B*-deleted transformants. We speculate that the pathogenicity of the “cyst germinated mycelia” was possibly reduced during cyst germination to invade the host. Therefore, targeting the PcSTT3B may be a promising strategy for the development of fungicides with a new mode of action.

In eukaryotes, the ER is a protein-folding processing factory. To ensure correct protein folding, the ER quality control (ERQC) recognizes the Glc_3_Man_9_GlcNAc_2_ structure on the NXS/T motif of the nascent polypeptides [52]. The unfolded protein response (UPR) assists in refolding the misfolded protein to avoid the escape of misfolded or intermediate proteins from the ER. The irreparable protein is degraded through the ER-related degradation (ERAD) system to ensure that only correctly folded proteins are transported into the GA [53,54]. The repression of *AfSTT3* activates the UPR in *A. fumigatus* [23]. *PcSTT3B* is required for *P. capsici* to cope with ER stress. Therefore, the deletion of the *PcSTT3B* gene might have damaged ERQC and then changes the biological phenotypes. Furthermore, the changes in the biological phenotypes of the *PcSTT3B*-deleted transformants are partially caused by the reduced N-glycoproteins.

In the prevention and control of *Phytophthora*, chemical control still plays an important role. However, due to the emergence of drug-resistant strains, there is an urgent need to explore new fungicide targets. Previous studies have reported that a novel inhibitor A378-0 in the Pharmaceuticals’ HitS DNA Encoding Library (DEL) was screened and characterized by using the mitogen activated protein kinases 1 (MoMps1) in *Magnaporthe oryzae* as bait, which inhibits the penetration and invasive growth of appressoria cells of rice blast fungus [55]. In this study, we have demonstrated that the PcSTT3B is necessary for the development, pathogenicity, and N-glycosylation of *P. capsici*, and can be considered a drug target for the control of *Phytophthora*. PcSTT3B serves as a baiting protein for the screening of small molecule compounds with novel chemical structures and good antibacterial activity in the DEL. These compounds can be used as a precursor for further design and development of fungicide compounds, as well as compounds that can be used as broad-spectrum fungicide combinations. This study provides a foundation for the effective prevention and control of various *Phytophthora* diseases, offering a new strategy for future research.

Overall, this study demonstrated that N-glycosylation has an indispensable and crucial role in *P. capsici*. However, the functions of key catalytic enzymes within the N-glycosylation are ambiguous. Hence, we identified and characterized the functions of a novel *PcSTT3B* gene, encoding an STT3 subunit in the OST, and analyzed its expression profile during the *P. capsici* life cycle. *PcSTT3B* was essential for vegetative growth, sporangial release rate, zoospore production, and virulence. PcSTT3B also contributed to the ER stress response and N-glycosylation. These results would enrich our knowledge of oomycete pathogenesis and consequently provide potential fungicide targets for oomycete control.

## 4. Materials and Methods

### 4.1. Pathogen and Plant

The *P. capsici* strain BYA5 was isolated from an infected pepper from the Gansu province of China in 2011. It served as the wild-type (WT) strain for gene editing to obtain transformants. The *PsSTT3B*-deleted transformants (B8, B146) were conducted by homology-directed repair (HDR)-mediated replacement of the *PcSTT3B* gene with an *NPTII* gene using CRISPR/Cas9 [30,31]. The transformant B-CK, which was transformed but failed to delete the *PcSTT3B* gene, was included as a control strain. The transformant CB151 was complemented by reintroducing *PcSTT3B* through the CRISPR/Cas9 mediated in situ complementation method [32]. All strains were routinely cultured on 10% V8 agar at 25 °C in darkness [56]. A susceptible pepper cultivar (Green Peak Ox Horn) was grown in a greenhouse at 27 ± 2 °C temperature, 80% relative humidity, 16 h-light/8 h-dark photoperiod.

### 4.2. Analysis of the PcSTT3B Protein

The *PcSTT3B* gene was cloned from *P. capsici* BYA5 with the primers listed in Appendix A. Other STT3 protein sequences from different species were acquired from the National Center for Biotechnology Information (NCBI, https://www.ncbi.nlm.nih.gov, accessed on 20 October 2022) database and the FungiDB database (https://fungidb.org/fungidb/app, accessed on 20 October 2022) [57]. The Mega 6.0 software was used to construct the phylogenetic tree [58]. MEME server (http://meme.nbcr.net, accessed on 20 October 2022) discovered motifs and domains in these proteins [59,60]. The prediction of transmembrane domains in the PcSTT3B protein was conducted using the TMHMM Server (http://www.cbs.dtu.dk/services/TMHMM/, accessed on 15 August 2022).

### 4.3. Vector Construction

The *PcSTT3B*-deleted transformants were generated using the CRISPR/Cas9 System [30,31]. The sgRNA for CRISPR/Cas9-mediated gene deletion was designed on the EuPaGDT server (http://grna.ctegd.uga.edu, accessed on 10 September 2021) and cloned into the pYF2.3G-ribo-sgRNA construct following a previously described protocol. This was named pYF2.3G-ribo-sgRNA-STT3B. One kb upstream and one kb downstream sequences of the *PcSTTB* gene were amplified and cloned into the pBSK+ donor vector and the replacement gene *NPTII* was inserted between them. This was named pBSK+-STT3B. 40-50 μg of DNA for each plasmid (pYF2.3G-ribo-sgRNA-STT3B, pBSK+-STT3B and pYF2-PsNLS-hSpCas9) were used for co-transformations to delete the *PcSTT3B* gene.

The entire *PcSTT3B* gene, ligated with its two flanking 1 kb fragments, was cloned into the pBSK+ donor vector, and named pBSK+-STT3B-C. The pYF2.3G-PcMuORP1-N (targeting the *NPTII* gene) [32], pBSK+-STT3B-C, and pYF-Cas9-EI [32] were co-transformed for gene complementation. The primers and sgRNA used in this study were listed in Appendix A.

### 4.4. P. capsici Transformation

The gene disruption and in situ complementation methods were performed using a CRISPR/Cas9-mediated gene replacement strategy [30,31,32]. Briefly, 2 days old *P. capsici* mycelial mats cultured in liquid pea broth medium were rinsed with ultra-pure water and suspended in 0.8 M mannitol with gentle shaking. After a 10 min incubation at 25 °C the mycelia were transferred to 20 mL of enzyme solution (0.4 M mannitol, 20 mM KCl, 20 mM MES, pH 5.7, 10 mM CaCl_2_, 0.3 g Lysing Enzymes from Trichoderma harzianum (Sigma, St Louis, MO, USA) and 0.3 g Cellulysin^®^ Cellulase (Calbiochem, San Diego, CA, USA) and incubated at 25 °C for approximately 30 min with gentle shaking (55–60 rpm). The mixture was then filtered through two layers of Miracloth (EMD Millipore Corp, Billerica, MA, USA), and the resulting protoplasts were collected by centrifugation at 530× *g* for 4 min in a 50 mL centrifuge tube.

After being washed with 30 mL of W5 solution (5 mM KCl, 125 mM CaCl_2_, 154 mM NaCl and 173 mM glucose), the protoplasts were resuspended in 5–10 mL of W5 solution and placed on ice for 30 min. The protoplasts were collected by centrifugation at 530× *g* for 5 min in a 50 mL centrifuge tube and resuspended at 10^6^–10^7^/mL in MMg solution (0.4 M mannitol, 15 mM MgCl_2_ and 4 mM MES, pH 5.7). The transformation itself was performed using 1 mL of protoplast suspension with 40–50 μg of DNA for each plasmid for co-transformations. The protoplast-plasmid mixtures were incubated on ice for 20 min, before 1740 μL of fresh PEG solution (40% PEG 4000 m/v, 0.2 M mannitol and 0.1 M CaCl_2_) was gradually added. The tubes were then shaken gently and incubated on ice for another 20 min before the protoplasts were mixed with 20 mL of pea broth (0.5 M mannitol, 9 mM CaCl_2_ and 20 mM CaCO_3_) and dark-incubated at 18 °C for 24 h to allow regeneration.

V8 agar containing 50 μg/mL G418 or 0.01 µg/mL oxathiapiprolin (DuPont Crop Protection, Wilmington, DE, USA) was used to select positive transformants [30,31,32]. PCR was used to detect the *PcSTT3B* gene replaced by *NPTII* in the *PcSTT3B*-deleted transformants. The *PcSTT3B*-deleted transformants and the *PcSTT3B*-complemented transformants were confirmed by RT-qPCR. The *PsSTT3B*-undeleted transformant was used as the control (B-CK) strain. The primers used in this study were listed in Appendix A.

### 4.5. Transcription Profile of the PcSTT3B Gene

Biological materials of *P. capsici* in the different development stages were collected as previously described, including mycelia (MY) from V8 agar, mycelia with sporangia (SP), zoospores (ZO), germinated cysts (CY), and infection stages (0, 1.5, 3, 6, 12, 24 and 48 h after inoculation on the pepper leaves) [61]. Total RNA was extracted from above biological materials of *P. capsici* using the SV Total RNA Isolation Kit (Promega, Beijing, China). RNA integrity was tested via agarose gel electrophoresis. After DNase I treatment, 1 µg of total RNA used for cDNA synthesis with the PrimeScript RT reagent Kit (Takara, Beijing, China) according to the manufacturer’s instruction.

RT-qPCR was performed in a qPCRsoft 3.4 system (qTower 2.2, Analytik Jena AG, Jena, Germany) using SYBR Premix Dimer Eraser Kit (CW Biotech, Beijing, China) under the following conditions: 95 °C for 2 min, 40 cycles of 95 °C for 10 s, 60 °C for 30 s to calculate cycle threshold (Ct) values. *Actin* and *WS21* genes were used as internal control [62]. Primers used for RT-qPCR in this study are listed in Appendix A online. The relative expression levels of genes were calculated using the 2^−ΔΔCt^ method [52]. Three biological replicates, each containing three technical replicates, were performed for each sample.

### 4.6. Phenotypic Characterization of the PsSTT3B-Deleted Transformants

To examine mycelial growth, cultural plugs (5 mm in diameter) were transferred onto V8 agar plates and incubated for 3 days, which had three replicates. Colony diameters were measured in two perpendicular dimensions.

To evaluate sporangium production, the culture was incubated at 25 °C in darkness on V8 agar plates for 3 days, and then in a 12 h-light/12 h-dark photoperiod for 5 days as previously described [63]. A microscope with 10× mirror magnification was used to count sporangia in the entire field of vision. Sporangia were observed either released zoospores or not. The sporangial release rate = the number of released sporangia/(the number of released sporangia + the number of unreleased sporangia).

To assess zoospore production, 10 mL of water was added to the plates, and then at 4 °C for 30 min and at 25 °C for 30 min. The zoospore suspensions were collected, and the concentrations were determined using a hemocytometer.

To evaluate cyst germination, the zoospore suspensions were shaken on a vortex for 1 min to form cysts and incubated at 25 °C for 3 to 8 h until most of the WT strain germinated. Cyst germination rates were calculated by counting the germinated cysts in the 100 individuals. Cysts were considered as germinated if the germ tube was longer than the cyst diameter.

To examine pathogenicity, the zoospore suspension was obtained as described above. Zoospore concentrations of all candidate strains were adjusted to 50,000 zoospores mL^−1^, and 10 µL were inoculated to the back side of pepper leaves. The inoculated leaves were incubated in chambers with 80% humidity at 25 °C in darkness. The diameter of the disease lesions was measured perpendicularly after 3 days. The symptoms were observed under ultraviolet light. Three independent experiments were conducted, and each consisted of 6 to 8 replicates. All three experiments showed similar results.

### 4.7. Tolerance of the PcSTT3B-Deleted Transformants to ER Stress Inducers

To evaluate *P. capsici* with misfunctioned *PcSTT3B* on the tolerance to ER stress inducers, fresh 5 × 5-mm hyphal plugs of the *PcSTT3B*-deleted transformants and the WT strain were inoculated on V8 agar plates amended with either 0.5 µg/mL TM or 6 mM DTT and incubated at 25 °C in darkness for 3 days with three replicates. The unamended V8 agar was used as a control. Colony diameters were measured with a fine crosshair. The inhibition rate was calculated as: (colony diameter without ER stress inducer—colony diameter with ER stress inducer)/(colony diameter without ER stress inducer—5 mm).

### 4.8. Western Blotting

The total proteins of *P. capsici* were extracted with the YT-015 Minute Total Protein Extraction Kit (Invent, Beijing, China). Protein content was measured using the Bradford (1976) method [64]. Total proteins were subjected to 10% SDS-PAGE gel for western blotting analysis via using the peroxidase-conjugated Concanavalin A (ConA-HRP) anti-body (Sigma-Aldrich, Shanghai, China). β-tubulin (Biodee, Beijing, China) was used as a control.

### 4.9. Data Statistics

All experiments were replicated three times, each producing similar results. The data were analyzed with the one-way analysis of variance (ANOVA) method followed by the Fisher’s least significant difference (LSD) test. Significance of means separation was determined based on *p* < 0.05 and *p* < 0.01.

## 5. Conclusions and Perspective

In summary, our study demonstrated that the N-glycosylation inhibitor TM, and ER stress inducer DTT hindered the growth of mycelia, sporangial release, and zoospore production in *P. capsici*, suggesting that N-glycosylation plays a crucial role in oomycete development. Moreover, *PcSTT3B*, the core subunit of OST, was found to be involved in pathogenicity, the ER stress response, and the N-glycosylation of *P. capsici*. This work sheds light on the molecular and biological functions of the *PcSTT3B* gene in *P. capsici* and enriches the understanding of the functional mechanisms and regulatory pathways of eukaryotic STT3B. Importantly, our results also contribute to filling the international research gap on the functions of eukaryotic STT3B in oomycetes.

## Figures and Tables

**Figure 1 ijms-24-07500-f001:**
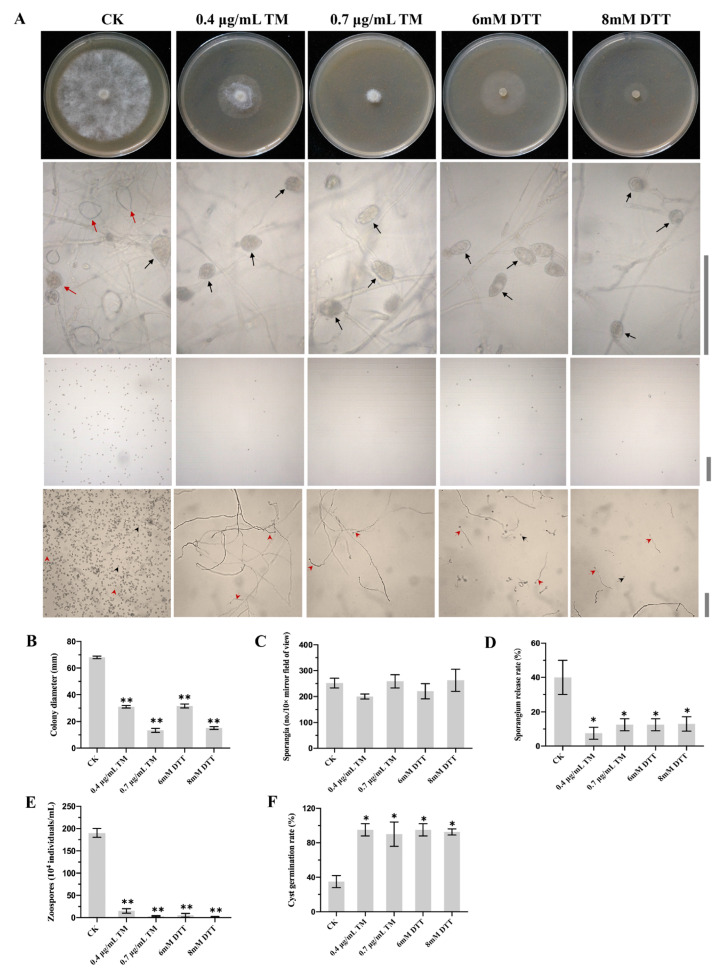
Effects of N-glycosylation inhibitor Tunicamycin (TM) and ER stress inducer dithiothreitol (DTT) on the development of *Phytophthora capsici*. (**A**) Mycelial growth on agar plates, microscopic visualization of sporangium production, zoospore release and cyst germination supplemented with TM (0.4 µg/mL or 0.7 µg/mL) or DTT (6 mM or 8 mM). Red solid line arrows represent released sporangia. Black solid line arrows represent unreleased sporangia. Red dashed arrows represent germinated cysts. Black dashed line arrows represent ungerminated cysts. Bar = 100 μm. (**B**) Colony diameters of different strains cultured for 3 days on V8 agar plates supplemented with TM or DTT. (**C**) Sporangium productions of all strains under a light microscope at 10× mirror magnification. (**D**) Sporangial release rates of different strains. (**E**) Zoospore productions of different strains. (**F**) Cyst germination rates of different strains. Data represent the mean ± standard deviations from three biological repeats. Asterisks indicate a significant difference compared to the untreated control (CK) (*, *p* < 0.05; **, *p* < 0.01).

**Figure 2 ijms-24-07500-f002:**
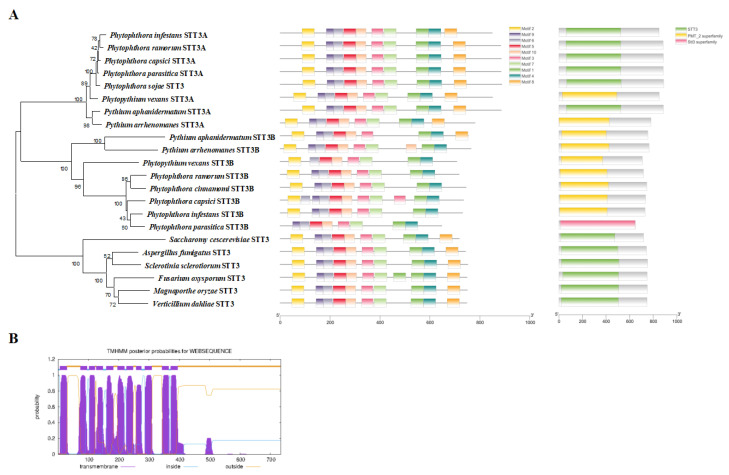
Sequence analyses of PcSTT3B protein. (**A**) Neighbor-joining phylogenic tree based on STT3 proteins of oomycetes and fungi. Bootstrap percentages indicate the occurrence probabilities of 1000 replicates. The scale bar represents 20% weighted sequence divergence as the scale of the evolutionary tree. The branch length reflects the evolutionary distances among STT3 proteins. Motifs and domains of STT3 proteins are in the oomycetes and fungi. (**B**) Transmembrane domain analysis of PcSTT3B using TMHMM.

**Figure 3 ijms-24-07500-f003:**
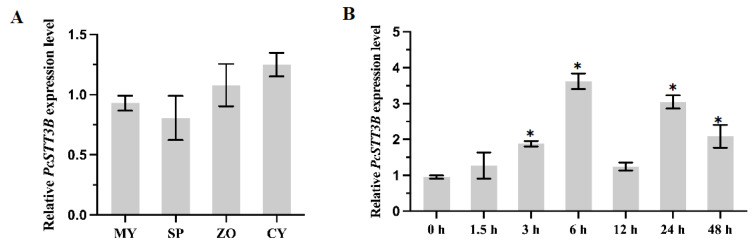
Expression patterns of the *PcSTT3B* gene at different stages of *P. capsici*. Relative expression levels of the *PcSTT3B* gene at the developmental (**A**) and post-infection (**B**) stages were measured by RT-qPCR. MY: mycelia. SP: sporangia. ZO: zoospores. CY: germinated cysts. Measurements were done at 0, 1.5, 3, 6, 12, 24, and 48 h post inoculation. Data represent the mean ± standard deviations from three biological repeats. Asterisks indicate significant differences from the MY or 0 h (*, *p* < 0.05).

**Figure 4 ijms-24-07500-f004:**
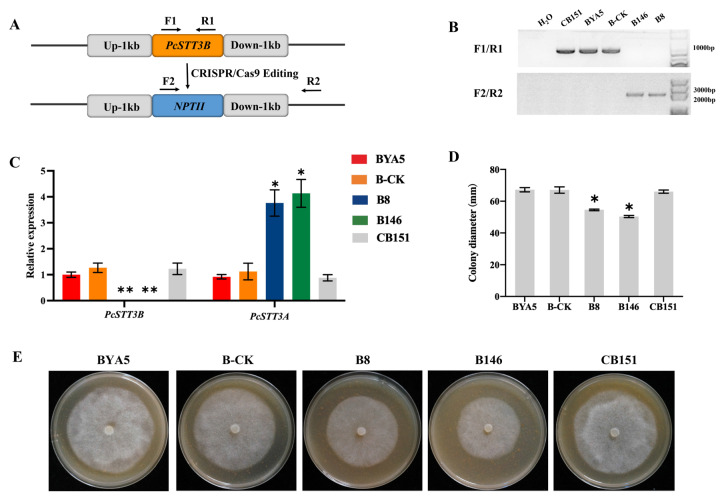
Roles of *PcSTT3B* to mycelial growth of *P. capsici*. (**A**) Diagram of HDR-mediated replacement of the *PcSTT3B* gene with the *NPTII* gene. The orange box represents the *PcSTT3B* gene, and the blue box represents the *NPTII* gene. (**B**) Verification of three *PcSTT3B*-deleted transformants using polymerase chain reaction (PCR). The Primers F1 and R1 were used to amplify the *PcSTT3B* gene. Primers F2 and R2 were used to amplify the *NPTII* gene. (**C**) Detection of *PcSTT3A* and *PcSTT3B* genes of different *P. capsici* strains were analyzed using reverse transcription quantitative real-time PCR (RT-qPCR). (**D**) Colony diameters of different *P. capsici* strains. (**E**) Colony morphology of *P. capsici* different strains cultured on the V8 agar for 3 days in darkness. Data represent the mean ± standard deviations from three biological repeats. Asterisks indicate significant deviations from the WT strain BYA5 (*, *p* < 0.05; **, *p* < 0.01).

**Figure 5 ijms-24-07500-f005:**
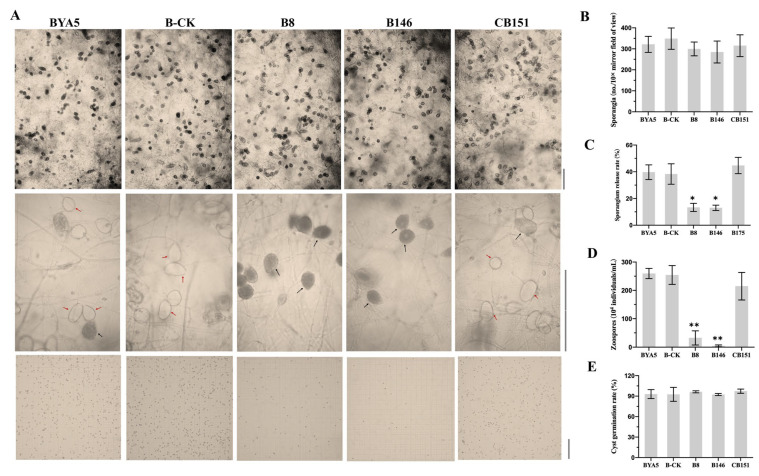
Effects of *PcSTT3B* on sporangia and zoospores in *P. capsici*. (**A**) Microscopic observation of sporangia, sporangial release, and zoospores. Red solid line arrows represent released sporangia. Black solid line arrows represent unreleased sporangia. Bar = 100 μm. (**B**) Sporangium production observed at 10× mirror magnification power. (**C**) Sporangial release rates of different strains. Sporangial release rate = the number of released sporangia/(the number of released sporangia + the number of unreleased sporangia). (**D**) Zoospores examined with a hemocytometer. (**E**) Cyst germination rates of different strains. Data represent the mean ± standard deviations from three biological repeats. Asterisks indicate significant differences from the Wild type BYA5 strain (*, *p* < 0.05; **, *p* < 0.01).

**Figure 6 ijms-24-07500-f006:**
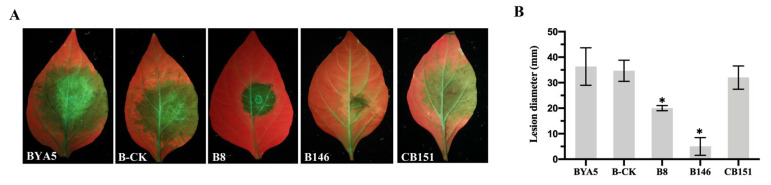
Effects of *PcSTT3B* on virulence in *P. capsici*. (**A**) Visualized lesion symptoms on peppers. (**B**) lesion diameters on peppers. Pepper leaves were inoculated with ∼100 zoospores and assessed after 3 days. All experiments were replicated three times, each had a similar result. Asterisks indicate significant deviations from the WT strain BYA5 strain (*, *p* < 0.05). Data represent the mean ± standard deviations from three biological repeats.

**Figure 7 ijms-24-07500-f007:**
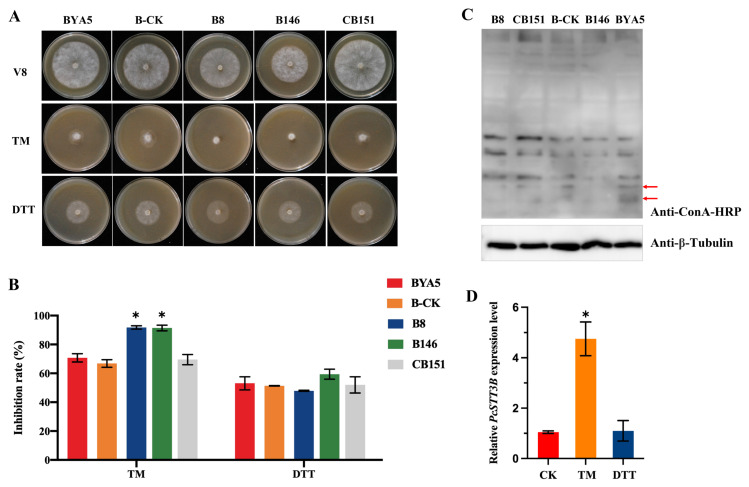
Roles of *PcSTT3B* in ER stress and N-glycosylation in *P. capsici*. (**A**) Mycelial growth assays of all strains on the V8 agar supplemented with 0.5 µg/mL TM or 6 mM DTT. TM, tunicamycin. DTT, dithiothreitol. (**B**) Mycelial inhibition rates under ER stress inducers. Colony diameters were measured 3 days after treatment. (**C**) Western blotting analysis for identifying glycoproteins. The total proteins of mycelia were extracted and analyzed by western blotting using an anti-ConA-HRP antibody. Red arrows represent certain mannosylated protein bands that slightly reduced in the *PcSTT3B*-deleted transformants compared to that of the BYA5 strain. β-tubulin was used as a loading control. (**D**) Relative *PcSTT3B* gene expression levels under ER stress inducers. The value of the un-treated control (CK) was set to 1 as a reference. Asterisks indicate significant differences from CK (*, *p* < 0.05). Data represent the mean ± standard deviations from three biological repeats.

## Data Availability

All the data generated or analyzed during this study were included in this article and its additional data files.

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
