# Peer review of "Characterization of PcSTT3B as a Key Oligosaccharyltransferase Subunit Involved in N-glycosylation and Its Role in Development and Pathogenicity of Phytophthora capsici"

_ijms, 2023, doi:10.3390/ijms24087500_

Round 1

Reviewer 1 Report

In the manuscript entitled “Oligosaccharyltransferase subunit PcSTT3B involves in the development, pathogenicity and N-glycosylation of Phytophthora capsici,” the authors demonstrated the phylogenic characterization of PcSTT3B protein, showed the correlation of PcSTT3B expression levels and different infection stages. The author further applied a homology-directed repair (HDR)-mediated replacement by CRISPR/Cas9 approach to investigate the role of PcSTT3B in response to ER stress and in the pathogenesis of infected plant. The data in this study were well organized, and depict a delicate functional demonstration of PcSTT3B in plant pathogenic oomycete. However, there are some concerns regarding the presentation of data analyzed, and the content of some graphs in the article. In particular, the following points should be considered:

Major points:

1. The authors showed that PcSTT3B-deleted transformants are more sensitive to ER stress inducer and display low glycoprotein content in the mycelia. However, effect of PcSTT3B depletion was enormous on the entire proteome. The authors should further show possible specific ER-stress-modulating protein(s) or virulence factor(s) which was glycosylated by PcSTT3B to more specifically address the pathogenic mechanism. Perhaps the differentially expressed proteins in Figure 7C could be characterized.

2. In Figure 4c, expression of PcSTT3A was elevated in PcSTT3B-deleted transformants B8 and B146, and showed a possible complementary expression. However, in Figure 7C, B146 still demonstrated a loss of glycoproteins pulled-down by ConA. Please provide possible explanation (with or without experiments) for these results.

3. Please describe how the finding of this study could be applied in agricultural industry?

Minor points:

1. For Figure 1A and Figure 5A, the authors may label the location of sporangial production (SP), zoospore release (ZO), and cyst germination (CY) with arrow heads in the photos to make a more clear indication.

2. Resolution of the bar graphs in Figure 1 and the labels in those graphs should be improved.

3. Resolution of figures and the font size in Figure 2 should be improved.

4. Resolution of the bar graphs in Figure 5 and the labels in those graphs should be improved.

Reviewer 2 Report

Dear Authors,

the presented topic is for a niche of researchers interested in plant pathogens and disease, but anyway in line with the aims of the Journal. The general opinion about the manuscript is good, including the design of the study, interesting data/outcomes reported by you, and the quality of presentation (figures and images are for the most clear and attractive).

Overall comments/suggestions

The title could be made more attractive, such as the keywords.

The introduction has to be implemented, with more useful information about Phytophtora capsici and about the role of investigated target genes.

As the same way, the discussion has to be implemented with other recent references, comparing data obtained in other similar studies, if present.

In material and methods section please better clarify the strains of P.capsici used and also the acronym forms. Furthermore in "Vector construction" and "P. capsici transformation" paragraphs has to be detailed, even if references are reported, in particular for that concerns CRISPR/Cas9 technology, a recent but complex methodology.

Minor and specific comments/suggestions

Lines 47-49: this sentence is incomplete, please rephrase it.

Lines 79-81: please specify the different concentrations used for TM and DTT.

Figure 2: please, improve the resolution and the dimension of the figure, in particular the one of the neighbor-joining phylogenic tree.

Line 124: RT-qPCR is the acronym for reverse transcription quantitative real-time PCR, that is the assay adopted in this study. Please remodulate this here and in other parts of the main text. In "Transcription Profile of PcSTT3B gene" paragraph please specify the quantity/quality of extracted RNA.

Lines 147-148: please, improve this expression "RT-qpCR showed that the expression of the PcSTT3B gene disappeared". For example in this way: "The expression of the PcSTT3B was not modulated and detected in RT-qpCR assay".

Please, better check teh reported references in accordance with the format required by "IJMS-MDPI".

Best regards.

M.T.

Reviewer 3 Report

I appreciate Cui et al. for reporting a relevant and interesting study entitled “Oligosaccharyltransferase subunit PcSTT3B involves in the development, pathogenicity and N-glycosylation of Phytophthora capsica”. The presentation and flow of the article are of good quality. I have a few concerns, which I request the authors clarify and modify the manuscript as needed.

1.     The manuscript should have a conclusion section to conclude the study.

2.     The authors should discuss the interpretation of Figure 1 in the manuscript. The discussion that is currently in the manuscript is not enough.

3.     ORF of the PcSTT3A and PcSTT3B should be provided at least as the supplementary figure.

4.     What is the outgroup selected for the phylogenetic tree?

5.  No mention of how the greenhouse experiment was carried out. How were the pepper leaves and the pathogen inoculated and infected? Please provide details.

6.     What is the rationale for selecting 1.5, 3, 6, 12, 24, and 48 hrs for pepper-pathogen infection? This reasoning is critical for the manuscript.

7.     Modify the y-axis of figure 3 to mention the relative expression of which gene.

8.     Figure 7 legend: (Line #209): Roles, not rules.

Round 2

Reviewer 3 Report

The authors have revised the manuscript satisfactorily.